# Efficient Zeroth-Order Federated Finetuning of Language Models on Resource-Constrained Devices

**Mohamed Aboelenien Ahmed**                                          *mohamed.ahmed3@kit.edu*
*Karlsruhe Institute of Technology*

**Kilian Pfeiffer**                                          *kilian.pfeiffer@kit.edu*
*Karlsruhe Institute of Technology*

**Ramin Khalili**                                          *ramin.khalili@huawei.com*
*Huawei Heisenberg Research Center (Munich), Germany*

**Heba Khdr**                                          *heba.khdr@kit.edu*
*Karlsruhe Institute of Technology*

**Jörg Henkel**                                          *henkel@kit.edu*
*Karlsruhe Institute of Technology*

**Reviewed on OpenReview:** *https://openreview.net/forum?id=nVmz9Q2l7L*

## Abstract

Federated Learning (FL) is a promising paradigm for finetuning Large Language Models (LLMs) across distributed data sources while preserving data privacy. However, finetuning such large models is challenging on edge devices due to its high resource demand. Zeroth-order Optimization (ZO) estimates gradients through finite-difference approximations, which rely on function evaluations under random perturbations of the model parameters. Consequently, ZO with task alignment provides a potential solution, allowing finetuning using only forward passes with inference-level memory requirements and low communication overhead, but it suffers from slow convergence and higher computational demand. In this paper, we propose a new ZO-based method that applies a more efficient technique to reduce the computational demand associated with using a large number of perturbations while preserving their convergence benefits. This is achieved by splitting the model into consecutive blocks and allocating a higher number of perturbations to the second block, enabling efficient reuse of intermediate activations to update the full network with fewer forward evaluations. Our evaluation on RoBERTa-large, OPT1.3B, LLaMa-3-3.2B models shows up to $3\times$ reduction in computation compared to the other ZO-based techniques, while retaining the memory and communication benefits over first-order federated learning techniques.

## 1 Introduction

Large Language Models (LLMs) are demonstrating high performance in various machine learning tasks, including next-token prediction and text classification (Brown et al., 2020; Touvron et al., 2023; Zhang et al., 2022; Wu et al., 2023; Zhang et al., 2022). Despite their popularity, these models are very costly to train in terms of required data, computation, and memory resources. The current trend is to pre-train large models for extended periods on highly specialized hardware using large collections of public data, followed by task-specific finetuning. In many real-world deployments, however, the data required for finetuning is distributed across multiple devices and cannot be centralized due to privacy and regulatory constraints. Federated learning (McMahan et al., 2017) provides a framework for collaborative fine-tuning across multiple devices, allowing the model to benefit from diverse, decentralized data while preserving user privacy.

Federated fine-tuning of LLMs remains a resource-intensive task, particularly in terms of memory usage and communication overhead, posing significant challenges for deployment on resource-constrained edge devices. Parameter-efficient fine-tuning methods, such as LoRA (Hu et al., 2022), partially alleviate these challenges by reducing both memory and communication costs compared to full-model federated updates. However, despite these reductions, LoRA-based federated training still incurs substantial memory overhead, as intermediate activations must be stored to perform backpropagation. Moreover, while communication is reduced relative to standard FL, the transmission of low-rank adaptation parameters across clients and the server continues to introduce non-negligible communication costs. For example, finetuning RoBERTa-large (Liu, 2019) using LoRA (Figure 3) on a dataset with context length 256 requires memory of about 4 GB, which is not feasible for most edge devices (Imteaj et al., 2022; Pfeiffer et al., 2023b). It also requires over 100 MBs of data to be uploaded to the server throughout the FL training process, increasing the convergence time, especially when the uplink transmission rates are low (e.g., in wireless environments).

Memory Efficient ZO (MeZO) (Malladi et al., 2023) showed that it is possible to finetune language models with an inference-like memory footprint. This memory improvement is achieved by using only forward passes and pseudo-random seeds for gradient computation. The transmission of seeds for perturbation regeneration can additionally reduce communication costs in distributed settings. This makes ZO appealing in resource-constrained federated learning settings, where memory and communication are often constrained.

However, these improvements come at the expense of noisy gradient updates that could reduce the accuracy and convergence speed of the training. This limitation naturally extends to zeroth-order federated learning, where existing methods Li et al. (2025); Fang et al. (2022) mitigate the issue by employing a large number of perturbations per training step. In practice, this means that the loss function is evaluated multiple times along different random directions in each iteration to obtain an averaged gradient estimate. While averaging across many perturbations reduces the variance of the estimated gradient and improves stability, it significantly increases the computational burden, as each perturbation requires at least one additional forward evaluation. Thus, there exists a trade-off between gradient quality and computational efficiency that current ZO-based FL methods struggle to balance.

In this paper, we propose Federated Split-Perturbation Zeroth-order Optimization (FedSPZO), which reduces the computational cost compared to state-of-the-art federated zeroth-order approaches in finetuning language models. At the same time, it retains the key advantages of zeroth-order optimization, including improved memory efficiency and reduced communication overhead compared to federated first-order methods. FedSPZO splits the network into two consecutive blocks and applies a larger number of perturbations to the second, significantly smaller block, leveraging more function evaluations to update the entire network while reusing intermediate outputs from the first block to reduce computational overhead. In addition, the server reconstructs exact clients' models in multi-step FL setting, so the model aggregation can be performed without clients sending the model parameters. We evaluate the performance of FedSPZO on finetuning different models and datasets and observe that FedSPZO reduces the computation overhead by up to $3\times$ and $2.8\times$ compared with Fang et al. (2022) and Li et al. (2025), while keeping the memory overhead in the same order as MeZO. Compared to LoRA, FedSPZO requires more computation, as expected, but reduces the memory footprint and achieves a reduction of more than three orders of magnitude in communication cost (upload overhead), making it a preferable choice in systems with constrained memory or communication resources.

Our main contributions are:

- We propose FedSPZO, a novel zeroth-order federated learning method that partitions the model at clients into two consecutive blocks and assigns a larger perturbation budget to the smaller, deeper block, resulting in improved computational efficiency.

- Our evaluation over different models (Liu, 2019; Zhang et al., 2022; Touvron et al., 2023) and datasets demonstrates a lower total computation than existing federated ZO baselines, with minor accuracy degradation relative to first-order backpropagation in FL.

- We conduct a comprehensive ablation study evaluating a no-splitting baseline, the effect of the second-block perturbation budget relative to the first, and independent block-wise gradient computation to analyze the effectiveness of the FedSPZO technique.

## 2 Related work

**Federated Learning:** In FL, the significant computational and communication burdens placed on participating devices remain a challenge. Several techniques have been proposed to mitigate this burden. To reduce communication overhead, methods such as compression (Thakker et al., 2019; Haddadpour et al., 2021) and sketched updates (Shi et al., 2020) have been explored. Other approaches address computational overhead by training only subsets of the neural network (NN), such as random subsets (Caldas et al., 2018) or a sliding window approach (Alam et al., 2022). Progressive training has been introduced to alleviate communication and computational costs through model growth (Wang et al., 2022) and successive layer training with freezing, which also reduces memory requirements (Pfeiffer et al., 2023a). However, the majority of these works assume training from scratch.

Fine-tuning of large language models through FL has also been considered (Babakniya et al., 2023; Qi et al., 2024; Zhang et al., 2024) using LoRA (Hu et al., 2022). However, such techniques are not always feasible due to memory constraints. First-order forward gradients Baydin et al. (2022) with forward-mode automatic differentiation were proposed for parameter-efficient finetuning of LLMs by Panchal et al. (2024); Xu et al. (2024); however, forward gradients are computationally expensive and require a larger memory footprint compared to ZO. Furthermore, Panchal et al. (2024); Xu et al. (2024) additionally involve the upload of trainable parameters in a multi-step per round setting.

**Zeroth-order Optimization:** ZO has been adopted in black-box optimization due to the lack of accessible gradient information (Nikolakakis et al., 2022; Cai et al., 2021). MeZO (Malladi et al., 2023) showed that the fine-tuning of LLMs is possible using ZO when utilizing task alignment with prompt-based tuning (Gao et al., 2021). The paper builds on ZO-SGD (Spall, 1992) using central finite difference to estimate the gradients without backpropagation and proposes a random seed trick to reduce the memory footprint by generating perturbation vectors on the fly, thus consuming the same memory as inference and allowing a reduced cost to store checkpoints.

In FL, FedZO (Fang et al., 2022) applied zeroth-order methods to estimate gradients, where clients send the complete model parameters to the server. BAFFLE (Feng et al., 2023) uses zeroth-order for gradient estimation, which requires sending a scalar per step iteration (in the FedSGD setting) and sending the complete model parameters per round. For Feng et al. (2023); Fang et al. (2022), it takes at least 20 perturbations per step to train small-scale vision models from scratch. Ling et al. (2024) studied the theoretical properties of finetuning language models using ZO. DecomFL (Li et al., 2025) utilizes the seed and model reconstruction tricks in MeZO to send only the seeds and gradient scalars for communication between the server and clients in FL instead of sending the model parameters. FedKseed (Qin et al., 2024) also sends scalar gradients and seeds to reduce communication costs while considering an FL system in which the server does not hold the global model. Scalar gradients are accumulated with respect to their corresponding seeds, where the number of seeds is restricted to enable faster reconstruction of the model at clients given multiple update steps, since the server does not hold the global model.

Existing ZO-based FL methods mainly focus on communication aspects while overlooking the computational overhead introduced by the large number of perturbations required for gradient estimation. FedSPZO differs by exploiting the sequential compositional structure of neural networks, enabling gradient estimation over disjoint parameter blocks with different perturbation budgets, thereby reducing per-round computation without degrading convergence behavior.

## 3 Methodology

### 3.1 Background on Zeroth-order Optimization

Zeroth-order Optimization estimates first-order gradient information through finite-difference approximations based solely on loss evaluations, avoiding explicit backpropagation. It reduces the memory footprint compared to first-order backpropagation, as it does not require the storage of the activations in memory. Furthermore, utilizing the seed trick proposed by Malladi et al. (2023), it is possible to maintain an inference-

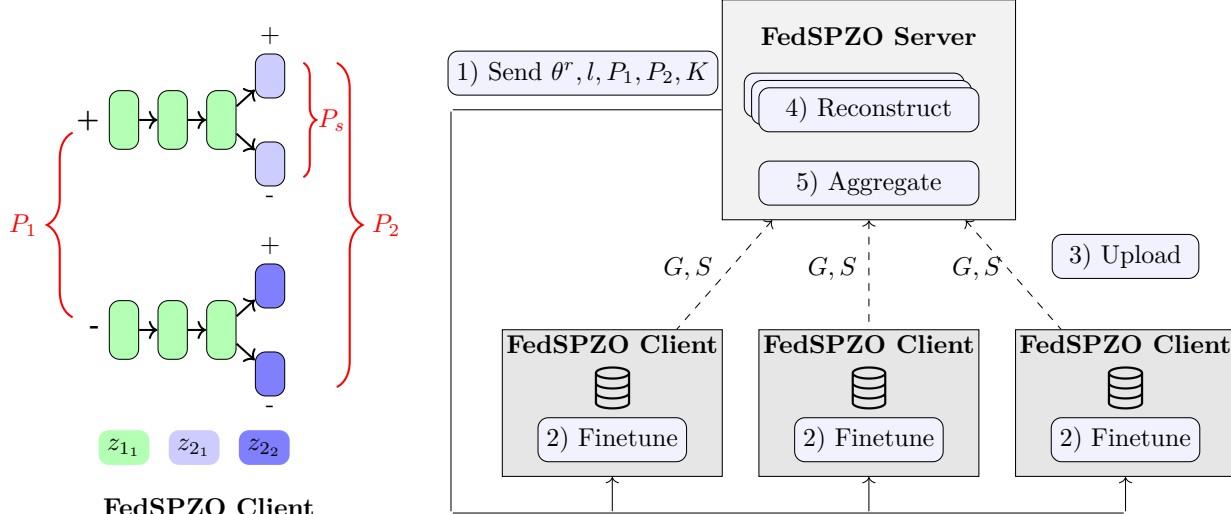

Figure 1: An overview of the proposed FedSPZO client and round design. Each client performs $P_1$ perturbations on the first parameter block $\theta_1$, and for both the positive and negative directions, executes $P_s$ inner perturbations on the second block $\theta_2$. The differences between output losses are used to estimate both blocks gradients. After $K$ local steps, the scalar gradients $G$ are then send to server to reconstruct the updated models of clients to be aggregated to start another round.

like memory footprint essential for edge devices participating in federated learning. Given model parameters $\theta$, loss $L$, and input batch $\mathcal{B}$, we define the *scalar* projected gradient (central difference between perturbed losses) $g$ as follows:

$$g = \frac{L(\theta + \epsilon z; \mathcal{B}) - L(\theta - \epsilon z; \mathcal{B})}{2\epsilon} \tag{1}$$

where $\epsilon$ is the perturbation scale, and $z$ is sampled from $\mathcal{N}(0, I_d)$, where $I_d$ is the identity matrix and $d$ is the number of parameters. By resetting the pseudo-random generator to seed $s$, each element of the perturbation vector $z$ can be generated to perturb the corresponding parameter in place. This perturbation is applied to the model before a forward pass and is reused to compute the gradient by multiplying the scalar $g$ and update parameters using the SGD rule. This approach eliminates the need to store pseudo-random perturbations in memory, as they can be regenerated on demand, thereby preserving an inference-like memory footprint. The previous gradient estimation described above corresponds to a single perturbation $P = 1$.

## 3.2 Federated Split-Perturbation Zeroth-order Optimization

We consider a standard FL setting consisting of a central server and a set of $\mathcal{C}$ clients. Each client holds a local dataset that is not shared with the server. The server coordinates the training process by broadcasting model parameters $\theta$ to a subset of participating clients in each communication round, where the selected devices locally train the model on their private data and return updates for aggregation.

Within this setting, we propose FedSPZO, a federated learning framework that utilizes ZO with a communication efficient design, where clients communicate only scalars to the server. Primarily, FedSPZO aims to address the computational challenges arising from ZO. It splits the network into two consecutive blocks while utilizing a larger number of perturbations for the second smaller block to accelerate computation and improve gradient estimation. Let $y$ be the output of a model, we define the first and second blocks $f_1$ and $f_2$ as $y = f_2(\theta_2; f_1(\theta_1; \mathcal{B}))$, where $\theta_1$ and $\theta_2$ are the parameters for each block, respectively.

---

**Algorithm 1** FedSPZO Server

---

**Require:** Clients $\mathcal{C}$, Number of sampled clients $m$, Model paramaters $\theta$, Learning rate $\mu$, Cuttof layer $l$, Number of perturbations $P_1$ and $P_2$

1: **for** $r$ in $1, \ldots, R$ **do**
2: $\quad$ $\mathcal{M} \leftarrow$ Sample $m$ clients from $\mathcal{C}$
3: $\quad$ **for** $c$ in $\mathcal{M}$ **do**
4: $\quad\quad$ $G_1^c, S_1^c, G_2^c, S_2^c \leftarrow$ **ClientTrain**$(\theta^r, l, P_1, P_2, K)$ {Algorithm 2}
5: $\quad$ **for** $c$ in $\mathcal{M}$ **do**
6: $\quad\quad$ $\theta^c \leftarrow$**Reconstruct**$(\theta^r, K, G_1^c, G_2^c, S_1^c, S_2^c, \mu, P_1, P_2)$
7: $\quad$ $\theta^{r+1} \leftarrow \frac{1}{m} \sum_{c \in \mathcal{M}} \theta^c$
8: **Reconstruct**$(\theta, K, G_1, G_2, S_1, S_2, \mu, P_1, P_2)$**:**
9: $\quad$ $\theta_1, \theta_2 \leftarrow$ Split $\theta$ at $l$
10: $\quad$ **for** $k$ in $1, \ldots, K$ **do**
11: $\quad\quad$ **for** $p$ in $1, \ldots, P_1$ **do**
12: $\quad\quad\quad$ $z_1 \sim \mathcal{N}(0, I_{|\theta|_1})$ with seed $S_1[k][p]$
13: $\quad\quad\quad$ $\theta_1 \leftarrow \theta_1 - \mu/P_1 \cdot z_1 \cdot G_1[k][p]$
14: $\quad\quad$ **for** $p$ in $1, \ldots, P_2$ **do**
15: $\quad\quad\quad$ $z_2 \sim \mathcal{N}(0, I_{|\theta|_2})$ with seed $S_2[k][p]$
16: $\quad\quad\quad$ $\theta_2 \leftarrow \theta_2 - \mu/P_2 \cdot z_2 \cdot G_2[k][p]$
17: $\quad$ $\theta \leftarrow \theta_1, \theta_2$
18: $\quad$ **return** $\theta$

---

### 3.2.1 FedSPZO Server

We first discuss the proposed FL design. In each round $r$, the server broadcasts the model parameters $\theta^r$, the cutoff layer $l$, and the number of perturbations for each block (i.e., $P_1$ and $P_2$). Each client trains the model for the $K$ steps on its local data, then uploads only the vectors of scalar gradients $G_1$ and $G_2$ and seeds $S_1$ and $S_2$ used in the training for each block. The server is responsible for reconstructing the exact model for each client after $K$ steps. It performs the same parameter updates as those performed on the client $c$ using the learning rate $\mu$, perturbation vectors (regenerated from seeds $S_1^c$ and $S_2^c$), and the corresponding scalar gradient (from $G_1$ and $G_2$) for each step $k$ (Algorithm 1).

Importantly, this reconstruction does not require carrying out any forward passes on the server or access to the training data of the devices. It introduces only a minor overhead associated with replaying $K$ local update steps of participating clients on the server, which is insignificant relative to training time. Finally, the server averages all the clients models and starts another round (see Figure 1).

### 3.2.2 FedSPZO Client

ZO requires a small learning rate $\mu$ for stable convergence, as the variance of its gradient estimator scales with the model dimension (Ghadimi & Lan, 2013; Nesterov & Spokoiny, 2017); nevertheless, recent evidence suggests that this dependence is effectively controlled by the low effective rank of the loss Hessian in deep networks (Malladi et al., 2023). Increasing the number of perturbations per step, $P$, reduces the noise in the gradient approximation, enabling faster convergence and more accurate gradient estimation. Yet, this increases the computation overhead per step (and the whole training process). To reduce this overhead, FedSPZO splits the network into two blocks of subsequent layers, utilizing perturbations $P_1$ and $P_2$. The parameters in the first block $\theta_1$ are perturbed in the first positive direction of $z_1$ (as shown in Figure 1) using seed $s_1$. Then a forward pass outputs $^+y_l$, where $^+y_l = f_1(\theta_1 + \epsilon z_1; \mathcal{B})$ and $l$ indicates the index of the last layer in $f_1$. The second block is perturbed in the two directions $\pm z_2$, where $z_2$ is sampled using seed $s_2$. For each direction, $f_2$ takes only $^+y_l$ and computes its loss. This process for $f_2$ is repeated $P_s$ times (i.e., $P_2 = 2P_1 \times P_s$). This process is repeated for the negative direction $(-z_1)$ of $f_1$ to output $^-y_l$. $f_2$ is then

perturbed using a different $\pm z_2$(s) than the ones used before, and the losses given $^-y_l$ are computed[1]. The zeroth-order gradient for $f_2$ is:

$$\nabla L(\theta_2; \mathcal{Y}) = \frac{1}{P_2} \sum_{j=1}^{P_2} g_{2_j} \cdot z_{2_j} \tag{2}$$

where $\mathcal{Y} = \{\pm y_l^1, \ldots, \pm y_l^{P_1}\}$ is the output of perturbed $f_1$ with $\{\pm z_{1_1}, \ldots, \pm z_{1_{P_1}}\}$ given $\mathcal{B}$, and $g_{2_j}$ is the difference between losses perturbed by $z_{2_j}$. For $f_1$, let $\mathrm{L}^+$ and $\mathrm{L}^-$ be two vectors containing all the losses from $f_2$ over a single perturbation over $f_1$ from each direction, the scalar projected gradient $g_1$ is calculated as follows:

$$g_1 = \frac{1}{4P_s^2} \sum_{k=1}^{2P_s} \sum_{l=1}^{2P_s} \left(\frac{\mathrm{L}^+(l) - \mathrm{L}^-(k)}{2\epsilon}\right) \tag{3}$$

and the gradient for $\theta_1$ with $P_1$ is:

$$\nabla L(\theta_1; \mathcal{B}) = \frac{1}{P_1} \sum_{j=1}^{P_1} g_{1_j} \cdot z_{1_j} \tag{4}$$

There are multiple benefits to using this estimation technique. Firstly, for the gradients of layers in the first block, it *reduces the noise introduced by the perturbations of the last layers.* This is achieved by averaging over multiple perturbations of the second block when computing the difference in loss values, thus allowing for a better estimate of the gradient along the sampled direction $z_1$.

For the second block, the advantage arises from the fact that *each perturbation uses the same input activations that originate from a single direction of perturbation $\theta_1$.* Furthermore, the increased number of perturbations for $f_2$ is made computationally efficient by *reusing the output of $f_1$.* Let the $\mathrm{fw}_{\mathrm{FLOPs}}^1$ and $\mathrm{fw}_{\mathrm{FLOPs}}^2$ be the inference FLOPs of the two blocks, the computation cost for a step is as follows:

$$\text{zo-fw}_{\mathrm{FLOPs}} = 2 \times \mathrm{fw}_{\mathrm{FLOPs}}^1 \times P_1 + 2 \times \mathrm{fw}_{\mathrm{FLOPs}}^2 \times P_2 \tag{5}$$

with the cost for perturbations and updates divided similarly, where $fw_{\mathrm{FLOPs}}^2 \ll fw_{\mathrm{FLOPs}}^1$.

In Algorithm 2, we provide the FL client training process. For each step $k$, the client samples a batch and starts perturbation iterations to compute the two scalar gradients, as discussed. The client then updates each block by regenerating $z$ of each parameter and multiplying it by $g$ to compute the gradient of the parameter and update it. This is done element-wise, as in MeZO, avoiding the need to explicitly store gradients for each parameter. The gradient scalars and seed are stored in $G_1, G_2$ and $S_1, S_2$, respectively, to be uploaded to the server after $K$ steps. The proposed method might add minimal memory overhead, requiring storage of only the activation of the cutoff layer ($y_l$). The peak memory for a training step is $d + \max(\max(\{y_1, \ldots, y_{l-1}\}),$ $y_l + \max(\{y_{1+1}, \ldots, y_N\}))$, where $y_i$ is the layer output at index $i$ and $N$ is the number of layers.

**FedSPZO Communication Extension:** For ease of readability, we presented the method such that the client sends both gradient scalars and seeds in a round. However, since the seeds $S_1$ and $S_2$ are derived using a pseudo-random seed, the server can independently regenerate them by sampling and send a different initial seed to each client to start with. This eliminates the need for the client to upload the seeds.

## 4 Results

In this section, we evaluate the effectiveness of our approach in terms of accuracy, memory usage, computational cost, and communication (upload) overhead. We first compare these metrics against state-of-the-art first-order federated learning methods based on backpropagation and then against zeroth-order methods in FL. Finally, we present an ablation study to analyze the contribution of each component of our approach.

---

[1]In the case where a parameter is shared between $f_1$ and $f_2$, we perturb this parameter only in $\theta_1$ such that it is included only in the forward pass of $f_2$ but not included in $\theta_2$.

---

**Algorithm 2** FedSPZO Client

---

**Require:** Model parameters $\theta$, Local steps $K$, Learning rate $\mu$, Cutoff layer $l$, Number of perturbations $P_1$, $P_s$, and $P_2$
1: Initialize $G_1[1\ldots K][1\ldots P_1], G_2[1\ldots K][1\ldots P_2]$
2: Initialize $S_1[1\ldots K][1\ldots P_1], S_2[1\ldots K][1\ldots P_2]$
3: $\theta_1, \theta_2 \leftarrow$ Split $\theta$ at $l$
4: **for** $k$ in $1, \ldots, K$ **do**
5:    $g_1, g_2, sl_1, sl_2 \leftarrow [], [], [], []$ {Initialize scalar grads and seeds lists per step }
6:    Sample $\mathcal{B}$
7:    **for** $p_i$ in $1, \ldots, P_1$ **do**
8:       Sample seed $s_1 \sim \mathcal{U}(\{0, 1, \ldots, 10^8\})$ and Append to $sl_1$
9:       Sample $2P_s$ seeds $\sim \mathcal{U}(\{0, 1, \ldots, 10^8\})$ and Append to $sl_2$
10:      Sample $z_1 \sim \mathcal{N}(0, I_{|\theta_1|})$ with seed $s_1$
11:      $g_2^+, \mathrm{L}^+ \leftarrow$ **Forward**$(\theta_1, \theta_2, z_1, \epsilon, \epsilon, sl_2[: P_s], P_s)$
12:      $g_2^-, \mathrm{L}^- \leftarrow$ **Forward**$(\theta_1, \theta_2, z_1, -\epsilon, \epsilon, sl_2[P_s :], P_s)$
13:      $g_1 \leftarrow \frac{1}{4P_s^2} \sum_{i=1}^{2P_s} \sum_{j=1}^{2P_s} \left(\frac{\mathrm{L}^+(j) - \mathrm{L}^-(i)}{2\epsilon}\right)$
14:      Append $g_1$ to $G_1[k]$ and $g_2^+, g_2^-$ to $G_2[k]$ {Store grads}
15:    $S_1[k], S_2[k] \leftarrow sl_1, sl_2$ {Store seeds}
16:    $\theta_1 \leftarrow \theta_1 - \mu \nabla L(\theta_1)$ {Equation (4)}
17:    $\theta_2 \leftarrow \theta_2 - \mu \nabla L(\theta_2)$ {Equation (2)}
18: **return** $G_1, S_1, G_2, S_2$ {Send to Server}
19: **Forward**$(\theta_1, \theta_2, z_1, \epsilon_1, \epsilon_2, sl, P_s)$: {One forward for $f_1$ and $P_s$ perturbations for $f_2$}
20:    Initialize $\mathrm{L}[1\ldots 2P_s]$ and $g \leftarrow []$
21:    $y_l \leftarrow f_1(\theta_1 + \epsilon_1 z_1 ; \mathcal{B})$
22:    **for** $p_j$ in $1, \ldots, P_s$ **do**
23:      Sample $z_2 \sim \mathcal{N}(0, I_{|\theta_2|})$ with seed $sl[p_j]$
24:      $loss^+, loss^- \leftarrow L(\theta_2 + \epsilon_2 z_2; y_l), L(\theta_2 - \epsilon_2 z_2; y_l)$
25:      Append $(loss^+ - loss^-)/2\epsilon_2$ to $g$
26:      $\mathrm{L}[p_j], \mathrm{L}[p_j + 1] \leftarrow loss^+, loss^-$
27:    **return** $g$, $\mathrm{L}$ {return scalar grads and list of evaluated losses}

---

## 4.1 Experimental Setup

We evaluate FedSPZO alongside other FL approaches with prompt-based (task-aligned) tuning (Gao et al., 2021), where downstream tasks are reformulated to match the pretraining objective without introducing a task-specific output layer while retaining the pretrained language modeling head. For the RoBERTa-large model (Liu, 2019), we conduct experiments on the SST-2 (Socher et al., 2013), RTE (Bowman et al., 2015), and WiC (Pilehvar & Camacho-Collados, 2019) datasets. We consider 100 clients for SST-2 and 20 clients for the other datasets due to their smaller sizes. To assess scalability to larger model sizes, we additionally evaluate OPT-1.3B (Zhang et al., 2022) on the MultiRC (Khashabi et al., 2018) (of context length 400) and SQuAD Rajpurkar et al. (2016) datasets, and LLaMA-3.2-3B (Touvron et al., 2023) (with float16) on BoolQ (Clark et al., 2019), with 20 clients for each task. In all experiments, 10% of clients are randomly sampled to participate in each FL round, and each selected client performs 20 local optimization steps per round. All approaches are trained until convergence using sufficiently long training horizons (up to 12K rounds for zeroth-order methods).

We measure communication cost as the total amount of data uploaded by clients to the server across rounds, accounting for all transmitted model updates or gradient scalars required by each method. For first-order methods, memory consumption includes model parameters, gradients, and intermediate activations. Activation memory is measured by traversing the PyTorch computation graph during backpropagation, capturing all tensors required for gradient computation. We quantify computational cost using the PyTorch profiler, which reports the total number of floating-point operations (FLOPs) executed by each method.

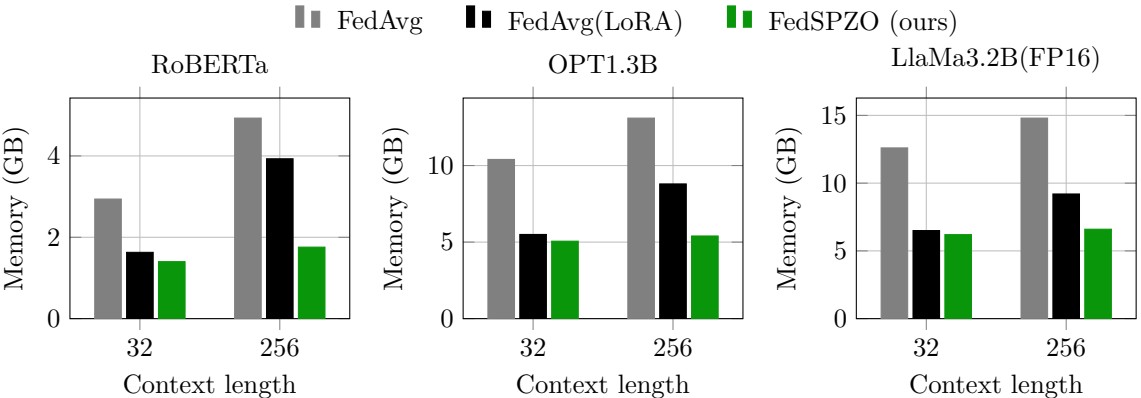

Figure 2: Memory footprint for training RoBERTa-large, OPT1.3B, LLaMA-3-3.2B (with float16) with context length 32 and 256 using first-order FedAvg w/o LoRA with backpropagation using batch size of 4, and FedSPZO using batch size 8.

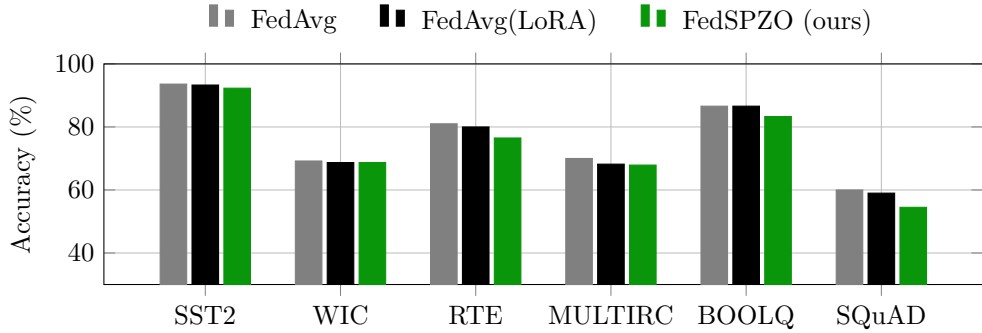

Figure 3: Accuracy comparison between FedSPZO and first-order backpropagation approaches over multiple models and datasets.

For FedSPZO, we split the model into two sub-networks such that $f_2$ starts from the last untied linear layer, while $f_1$ comprises all preceding layers, including the embedding layer and attention blocks. We set $P_1 = 2$ and $P_s = 2$, resulting in $P_2 = 8$ (check Section 4.5 for more discussion).

## 4.2 Comparison with First-order Methods

We compare FedSPZO against vanilla FedAvg (McMahan et al., 2017) and federated fine-tuning with LoRA (Hu et al., 2022), denoted as FedAvg(LoRA). For FedAvg(LoRA), we use a LoRA rank of $r = 8$ and a scaling factor of $\alpha = 16$, applied to the query and value projections of the attention layers.

**Memory Evaluation:** Figure 3 provides a comparison of the memory footprint of finetuning on RoBERTa-large and OPT1.3B using two different context lengths of 32 and 256. For FedAvg and FedAvg(LoRA), we consider vanilla SGD with no optimizer states (see also Appendix A for further discussion with AdamW (Loshchilov & Hutter, 2019)). FedAvg consumes up to 4.9 GB, 13.1 GB, and 14.8 GB over RoBERTa-large, OPT1.3B, and LLaMA, respectively. LoRA with backpropagation achieves competitive memory savings in the small context-length scenario; however, it suffers in the long context-length scenario as the cost of intermediate activations increases, requiring 3.9 GB, 8.8 GB, and 9.2 GB of memory for long context-length over the three models. For FedSPZO, it consumes a peak memory in the long context-length of only 1.75 GB, 5.4 GB, and 6.6 GB on the three models, while using a larger batch size. These show that first-order backpropagation methods require much larger memory as they store activations and gradients, making them impractical for training on edge devices.

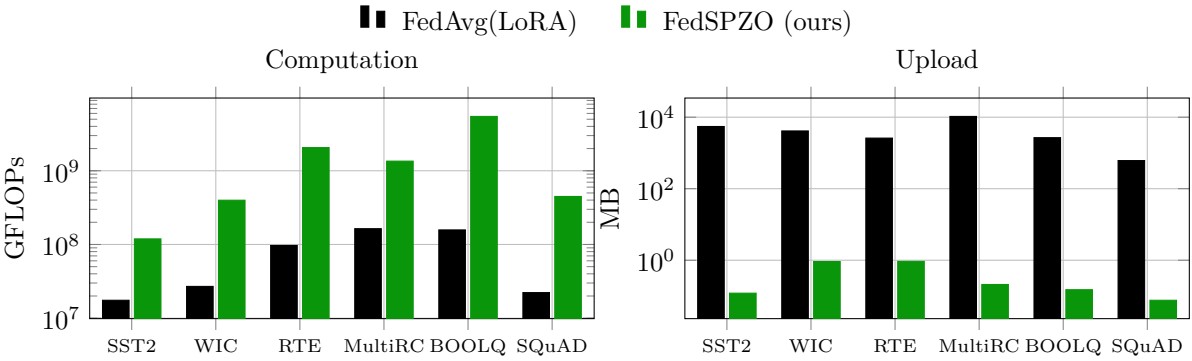

Figure 4: Computation and upload of finetuning FedSPZO and FedAvg(LoRA) over multiple settings.

**Accuracy:** In terms of accuracy, FedSPZO achieves results comparable to FedAvg(LoRA), with only a small decrease of 1% and 4% on RoBERTa over SST2 and RTE respectively, while showing no loss in accuracy over the WIC dataset. On OPT1.3B, FedSPZO shows a decrease of 0.4% and 7% over MultiRC and SQuAD (f1-score) while recording a decrease in accuracy on LLaMA of 3.8% over BOOLQ dataset. Both methods, however, remain slightly behind FedAvg, which serves as an upper bound. For FedSPZO, this is 3.9% on average, indicating a minor gap compared to FedAvg. This shows that FedSPZO (and ZO in general) do not lead to large degradation in accuracy in the evaluated finetuning scenarios.

**Computation and Communication:** Figure 4 provides a comparison of the computation and communication (upload volume) costs of FedSPZO and FedAvg(LoRA). We observe that FedSPZO uses, on average, $17\times$ higher total computation than FedAvg(LoRA) across the evaluated settings, primarily due to the faster convergence of FedAvg(LoRA). In terms of upload, while LoRA uploads only low-rank modules instead of full model parameters, it falls behind FedSPZO, which transmits only scalar values, by more than 3 orders of magnitude.

In summary, FedSPZO is well suited for scenarios with tight memory and communication budgets; however, methods such as LoRA remain more effective when such constraints do not exist.

### 4.3 Comparison with zeroth-order Methods

We compare our FedSPZO with zeroth-order DecomFL (Li et al., 2025) and FedZO (Fang et al., 2022). Both approaches utilize the forward difference for zeroth-order gradient estimation. Forward difference requires less computation compared to central difference since the network is only perturbed in one direction, thus conducting only $P + 1$ forward passes. For DecomFL, we set $P = 10$ equal to the original paper setting. For FedZO, the paper considered training from scratch while using high $P$ (e.g. 20 and more); however, we found this number is not required for the fine-tuning scenario. We did a grid search and selected $P = 5$ for FedZO as it provides a better option. In all zeroth-order optimization experiments, we disable the effect of train-time behaviors in layers, e.g., dropout, ensuring that these layers operate in inference mode during local updates. This setting eliminates additional stochasticity introduced by trainable layers, which can otherwise degrade gradient estimation quality, leading to reduced accuracy and slower convergence. While prior methods such as DecomFL did not explicitly adopt this configuration, we applied the same adjustment to ensure a consistent comparison across all methods. FedKSeed (Qin et al., 2024) uses central difference and relies on a finite pool of seeds from the server for gradient estimation on the clients' side, which can degrade model utility when the number of seeds is limited. The finite seed pool design stems from the assumption that the server does not retain a copy of the model, and as noted in Qin et al. (2024), it can lead to slower convergence compared to FedZO. The finite pool approach is considered orthogonal to ours in settings where the server does not have access to the global model. In Section 4.5, we also compare with different numbers of perturbation cases using central difference, which can be considered as an upper bound for FedKSeed in such a case.

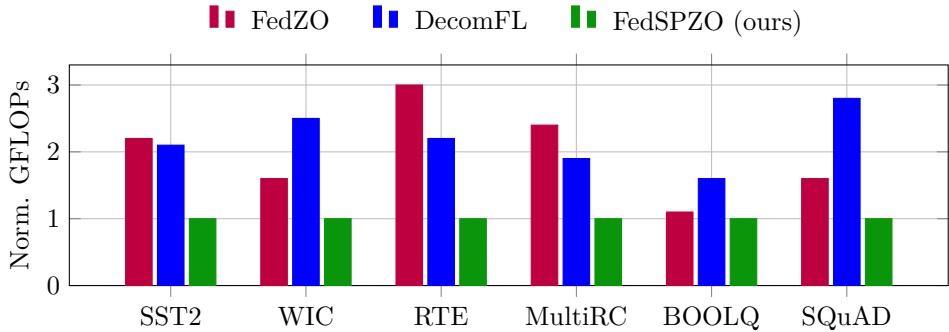

Figure 5: Total normalized GFLOPs comparison across ZO-FL training approaches, where all results are normalized to FedSPZO (ours). The results show that our approach achieves higher computational efficiency, requiring fewer total GFLOPs than other ZO-FL methods.

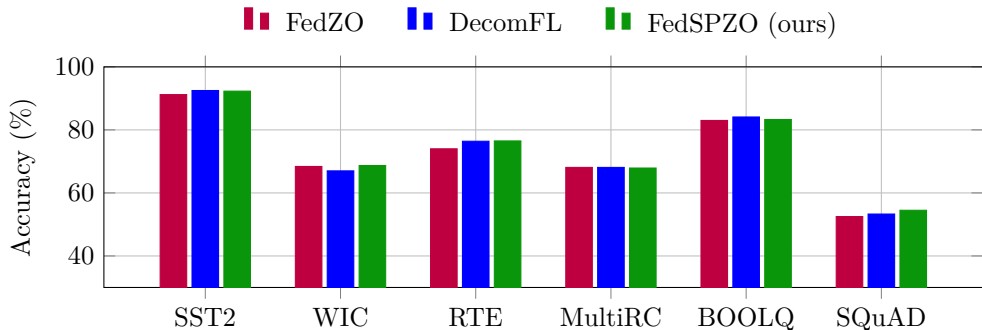

Figure 6: Accuracy comparison between FedSPZO and other ZO-FL approaches over different settings.

**Computation:** We compare the computation that is required for FedSPZO to reach the lowest recorded loss for the other DecomFL and FedZO. The results are illustrated in Figure 5. DecomFL uses $2.1\times$, $2.5\times$, $2.2\times$ GFLOPs compared to FedSPZO on SST2, WIC and RTE using the RoBERTa model, respectively. Further, it incurs $1.9\times$ and $2.8\times$ total GFLOPs on the MultiRC and SQuAD datasets when using the OPT-1.3B model, and $1.6\times$ on the BoolQ dataset when using LLaMA compared to FedSPZO. FedSPZO and DecomFL, given the larger number of perturbations used in DecomFL, converge in a similar number of rounds; however, FedSPZO provides much lower computation efficiency per round and thus in total. For FedZO, it uses $2.2\times$, $1.6\times$, and $3\times$ GFLOPs compared to FedSPZO on SST2, WIC, and RTE using the RoBERTa model, while it incurs $2.4\times$ and $1.6\times$ over MultiRC and SQuAD using OPT1.3B and $1.1\times$ over BOOLQ using LLaMA. These gains are attributed both to the improved gradient estimation achieved by using the central difference scheme and to the computational efficiency of the split perturbation technique.

**Accuracy:** Figure 6 shows the maximum accuracy achieved by DecomFL, FedZO, and FedSPZO respectively. All approaches record similar accuracy, which, as discussed before, accounts for a small degradation in comparison with first-order backpropagation approaches.

**Memory:** We consider that both FedZO and DecomFL can directly apply the memory reduction seed reconstruction trick proposed by MeZO (Malladi et al., 2023), as they do not require activation reuse and can therefore be used to train large models. Thus, we directly mention the memory overhead of FedSPZO compared to MeZO. FedSPZO only requires saving the output activation before the second block. Subsequently, FedSPZO adds only additional peak memory of 1 MB and 8 MB on top of 1.4 GB and 1.75 GB over the two context lengths for RoBERTa-large (i.e., 0.07% and 0.44% increase compared to MeZO) and 2 MB and 16 MB on top of 5.1 GB and 5.4 GB (i.e. 0.04% and 0.3% increase) for OPT1.3B.

**Communication:** FedZO sends the complete network parameters per round; therefore, the total upload in the FL training process is on the order of TB. In the simple implementation that we provided for FedSPZO, the client uploads $2K$ scalar gradients and $P_1K$ and $P_2K$ seeds. As discussed, since the server sends the starting pseudo seed generator, we record FedSPZO, where only the scalar gradients are sent. Similarly, DecomFL also sends only the gradient scalars. Compared to DecomFL, FedSPZO sends gradient scalars for $\theta_1$ and $\theta_2$ instead of for $\theta$. We therefore observe, in some cases, a small but overall negligible increase in upload overhead of our approach compared to DecomFL.

### 4.4 Training time discussion

To provide a clearer picture of the practical efficiency of FedSPZO, we consider a realistic system-level model of training time. Specifically, in round , where clients participate in parallel, we define the training time as the time taken by one client to complete its local computation and upload. We also neglect the server-side aggregation time, assuming it is negligible compared to client operations (Yang et al., 2020). For estimation, we assume an upload speed of 2 MB/s and a Jetson Orin Nano GPU with 1280 GFLOPS peak performance as a representation for an edge device. Under this setup, for training a RoBERTa-based model on the SST-2 dataset with 10 clients per round, the total training times for the federated process are: 0.45 hours for FedAvg(LoRA) and 2.6 hours FedSPZO. Therefore, FedSPZO is still behind parameter-efficient (LoRA) first-order methods. However, as mentioned, these approaches have a larger memory footprint, which is a scarce resource at edge devices. In comparison to zeroth-order methods, DecomFL is estimated to take 6 hours in total, while FedZO is estimated at 166 hours since it requires sending the whole model parameters in addition to the extra computation cost to FedSPZO. This magnitude becomes ever more important for more computational tasks; for example, DecomFL on WIC will use about 81 hours compared to 31 hours by FedSPZO, which leads to savings on the order of days in time and energy. These comparisons further demonstrate the practical efficiency of FedSPZO in zeroth-order federated training.

To summarize, FedSPZO offers a good trade-off between resource requirements and accuracy and thus can be used as an alternative to LoRA in scenarios with constrained memory and limited communication bandwidth, such as IoT devices connected to the server over wireless links.

### 4.5 Ablation Study

**No splitting:** To further assess the impact of the model splitting that our FedSPZO adopts, we compare it with training the network without any splitting while keeping all other design choices the same, i.e., applying the central difference and using the same FL round design as in FedSPZO. We consider the default setting for training the network using $P = 1$ and a higher number of perturbations, that is, $P = 4$. We use $P_1 = 2$ and $P_2 = 8$ as the setting for FedSPZO. In terms of accuracy, as shown in Figure 7, FedSPZO and the higher perturbation of $P = 4$ provides similar accuracies while outperforming the low perturbation scenario of $P = 1$. This especially appears in the case of the RTE dataset with RoBERTa, showing the importance of utilizing a higher number of perturbations for better gradient estimation.

For total computation, we begin our discussion with the $P = 4$ setting. In this configuration, it requires $1.3\times$ and $1.6\times$ the FLOPs of FedSPZO on the SST2 and RTE datasets, respectively, using RoBERTa. For MultiRC, the computation increases to $1.7\times$. These results highlight the computational efficiency of the proposed splitting strategy while maintaining the accuracy benefits associated with using a higher number of perturbations. The $P = 1$ scenario record $1.06\times$ and $2.1\times$ computation to FedSPZO were used over SST2 and RTE with RoBERTa, while using $1.3\times$ over MultiRC with OPT1.3b due to slower convergence. It is important to note that the single perturbation setting can be accuracy-limited in some scenarios, highlighting both aspects of computational efficiency and quality.

**Number of perturbations:** FedSPZO introduces an additional hyperparameter, denoted as $P_s$, into the conventional ZO training framework. Recall that $P_s$ controls the scaling between the two perturbation budgets by determining $P_2$ as a function of $P_1$. To analyze the impact of this design choice and guide configuration selection, we report results in Figure 8 for $P_s \in \{1, 2, 4\}$ under two base settings, $P_1 \in \{1, 2\}$.

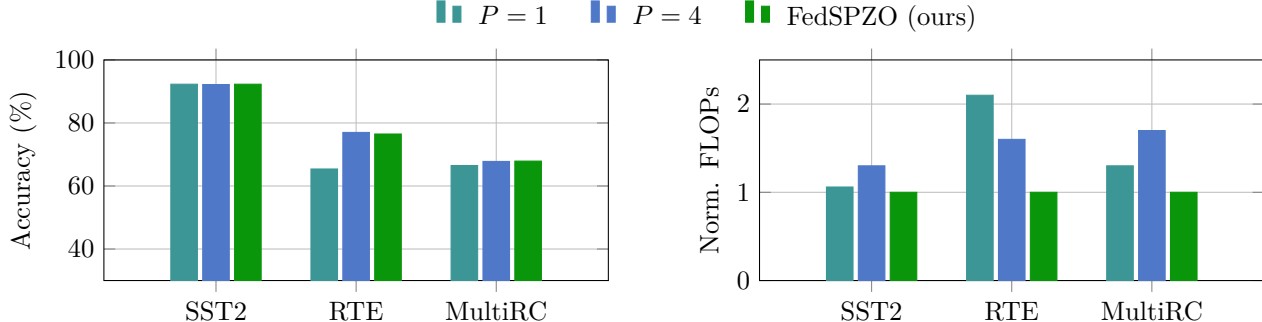

Figure 7: Accuracy and computation comparison between FedSPZO and the no-split scenarios using $P = 1$ and $P = 4$.

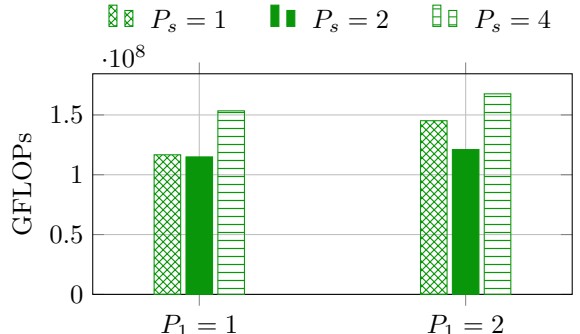

Figure 8: Effect of perturbation scaling $P_s$ on total computation. Results on RoBERTa over SST2 for different $P_1$ settings.

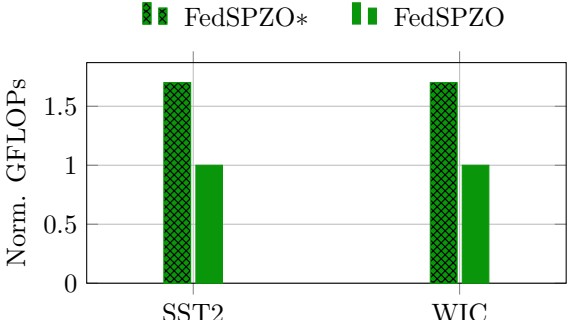

Figure 9: Evaluation of FedSPZO against a two-block zeroth-order baseline with $P = 2$ per group. The baseline estimates each block's gradient independently, leading to higher total computation.

Increasing $P_s$ improves convergence speed but also increases the computation cost per optimization step. As observed in the RoBERTa experiments on SST2, for both $P_1$ settings, increasing $P_s$ to 4 results in higher total computation compared to smaller $P_s$ values. This behavior arises because the marginal improvement in gradient estimation quality (with respect to $f_1$) becomes less significant relative to the additional computational overhead introduced by applying larger perturbation budgets to $f_2$. In contrast, $P_s = 2$ consistently achieves a more favorable trade-off, yielding slightly better convergence behavior than $P_s = 1$ without incurring excessive computational cost. Furthermore, all $P_s$ settings reach similar final accuracy, indicating no accuracy drop from different perturbation scaling choices in that evaluation setting.

**Cross perturbation:** To further analyze the effectiveness of the proposed FedSPZO technique, we introduce a baseline that uses the same network split (i.e., two parameter groups), where the gradient of each group is estimated independently. In this baseline, we first perturb the parameters of the first group, execute forward passes through the entire network, and compute the corresponding gradient. Next, we rerun a forward pass with the first group unperturbed to cache its activations, perturb the second group, and perform forward passes to estimate the second group's gradient. Finally, each parameter group is updated using its independently computed gradient.

It is important to emphasize that this baseline does not employ the tree-structured design of FedSPZO. Compared to FedSPZO: (i) the first block requires a larger number of forward passes (even with activation reuse), (ii) it does not benefit from additional function evaluations induced by perturbations applied to the second block, (iii) the second block performs a subset of forward passes without its own perturbations, and (iv) the gradients of the two blocks are computed independently, without cross-group perturbation

interactions. We evaluate this baseline using the RoBERTa model on the SST-2 and WIC datasets, with $P = 2$ perturbations per group as shown in Figure 9. The results show that this baseline consistently underperforms FedSPZO, incurring $1.7\times$ and $1.7\times$ computational costs of FedSPZO on SST-2 and WIC, respectively. These results further highlight the superior efficiency and effectiveness of the FedSPZO strategy.

## 5 Conclusion

In this work, we propose FedSPZO, a computation efficient ZO optimization framework for federated fine-tuning, and we studied how ZO can be efficiently adopted for the federated finetuning of LLMs, along with the potential gains it offers in reducing memory and communication overhead. Our experimental results demonstrate substantial computational efficiency, achieving an average reduction of $2\times$ compared to existing state-of-the-art zeroth-order FL methods. When compared to first-order FL with backpropagation, FedSPZO offers significant advantages in memory efficiency and reduced upload overhead, but exhibits lower computational efficiency as a limitation. For future work, we plan to explore zeroth-order on inference-only accelerators with lower precision. This direction aims to bridge the computational efficiency gap with backpropagation methods, further enhancing the practicality of FedSPZO in resource-constrained environments.

## Acknowledgments

This work was partially funded by the Federal Ministry of Research, Technology, and Space (BMFTR) as part of the DI-EDAI project (grant: 16ME0990K)

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

## A  Comparison with first-order with adaptive optimizers

In this section, we further discuss and compare first-order adaptive optimizers such as AdamW (Loshchilov & Hutter, 2019). AdamW maintains first- and second-moment estimates for each parameter, resulting in an optimizer state of $2\times$ the size of the trainable parameters on top of vanilla first-order SGD. This significantly increases the overall memory footprint compared to vanilla first-order SGD, making it less practical in federated settings such as FedAvg, especially in the absence of parameter-efficient techniques like LoRA as shown in Figure 10. To provide a more comprehensive comparison, we include additional results in Figure 11 for FedAvg(LoRA) trained with AdamW. denoted as FedAvg(LoRA)-AdamW. FedAvg(LoRA)-AdamW improves communication and computational efficiency relative to standard FedAvg(LoRA) used in our primary comparisons, with a minor increase in memory consumption. Furthermore, similar trends are observed when comparing against FedSPZO. FedSPZO offers clear advantages in terms of communication efficiency and memory usage, whereas FedAvg(LoRA)-AdamW yields improvements in computational efficiency and model accuracy.

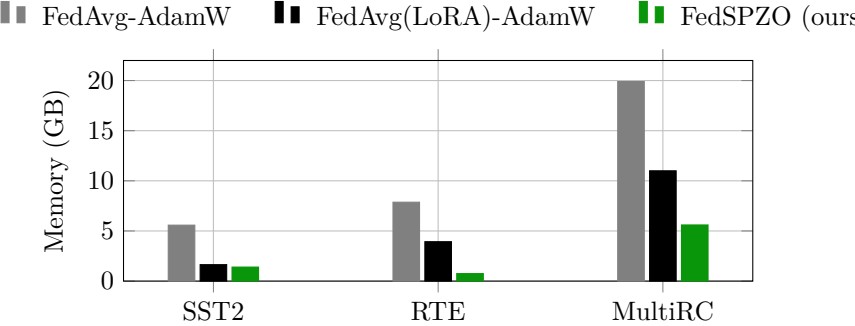

Figure 10: Peak Memory comparison between FedAVG and FedAVG(LoRA) using AdamW and FedSPZO on SST2 and RTE datasets (with RoBERTa-large) and MultiRC (with OPT1.3B and 400 context length).

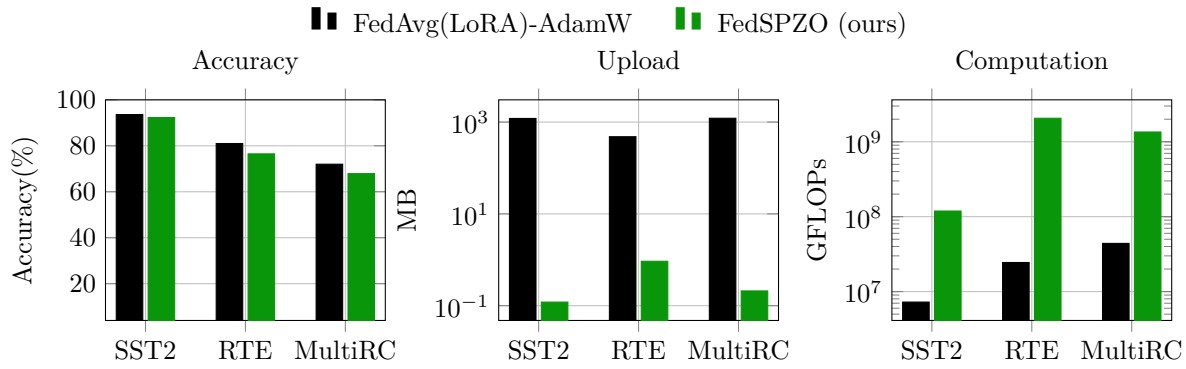

Figure 11: Accuracy, upload and computation comparison between FedAVG(LoRA) using AdamW and FedSPZO.

## B  LoRA and QLoRA with FedSPZO

In this section, we demonstrate that FedSPZO is fully compatible with parameter-efficient techniques such as LoRA. Moreover, we show that FedSPZO can be naturally combined with state-of-the-art quantization-based approaches, such as QLoRA (Dettmers et al., 2023). In Figure 12, we report accuracy results for FedSPZO(LoRA), FedSPZO(QLoRA), and FedSPZO with full parameters for reference, evaluated on the SST-2 and MultiRC datasets with RoBERTa and OPT models. The results indicate that LoRA and QLoRA are orthogonal to FedSPZO. Importantly, both variants exhibit only minor accuracy degradation, consistent

with the typical performance trade-offs observed when applying these techniques in first-order full-parameter fine-tuning.

It is worth noting, however, that when LoRA (without base-model quantization) is combined with memory-efficient zeroth-order approaches based on gradient reconstruction (including FedSPZO), it does not yield additional memory savings compared to full-parameter training. Similarly, in terms of upload, the advantage of transmitting only scalar values remains unchanged and is not further reduced by incorporating LoRA.

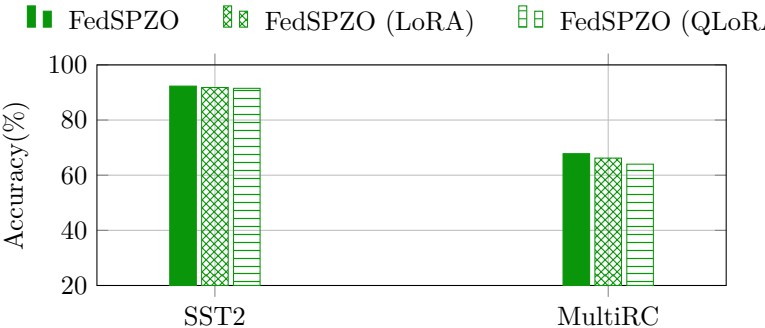

Figure 12: Accuracy comparison of FedSPZO, FedSPZO(LoRA), and FedSPZO(QLoRA). LoRA-based variants integrate seamlessly with FedSPZO and exhibit only minor accuracy degradation.

## C   Analysis on split placement

We investigate the impact of the split point on both computational cost and model performance under a fixed perturbation budget (i.e., $P_1 = 2$ and $P_2 = 8$). We observe that moving the split point earlier in the network (i.e., after the 17th, 21st, 23rd block out of 26 blocks of RoBERTa and OPT1.3B) as shown in Figure 13. This change can lead to an increase in computational cost compared to splitting at the final layer since a larger portion of the model must be executed for each perturbation in the zeroth-order gradient estimation. From an accuracy perspective, however, all four split configurations achieve comparable performance, with only a slight degradation observed for the 21st-block split on MultiRC dataset with OPT1.3B.

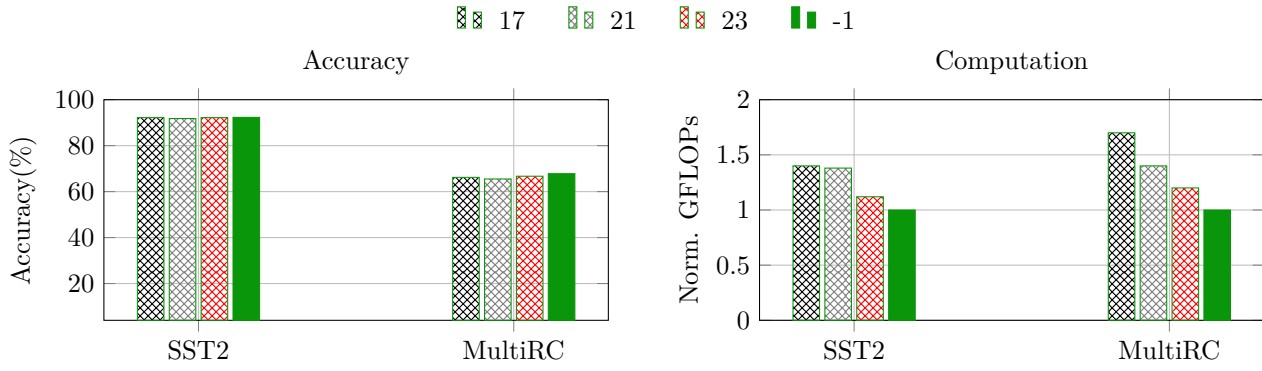

Figure 13: Accuracy and Normalized computation effect for different split point $l \in \{17, 21, 23-1\}$. (17/21/23 blocks; $l = -1$: final untied linear layer.

