# OpenReview forum: "Efficient Zeroth-Order Federated Finetuning of Language Models on Resource-Constrained Devices"
_TMLR — Accepted by TMLR_

### Review · Reviewer_3JQM · 2026-03-01

**Summary Of Contributions:**

The paper proposes FedSPZO, which splits a neural network into two blocks and applies more zeroth-order perturbations to the smaller tail block while reusing cached activations from the larger front block, cutting computation in federated ZO finetuning by up to 3x. Communication is reduced to scalar gradients and seeds.

The evaluation across memory, compute, communication, and accuracy is appreciated and gives a clear picture of where this method fits.

That said, I have a number of concerns. The split point is always fixed at the last linear layer and never varied (Section 4.1, "f2 starts from the last untied linear layer"), which feels like a pretty big gap since this is arguably the most important desig in the method. There's no analysis of what happens if you put the cut a few transformer layers earlier, so it's hard to know if the current choice is optimal or just convenient. On the theory side, the gradient estimator for theta_1 (Eq. 3) has this cross-product structure over losses from both blocks, and it's not clear whether this is unbiased or what its variance looks like compared to vanilla ZO with the same forward pass budget. The paper doesn't attempt any formal analysis here, which makes it hard to understand why the method works beyond intuition.

**Audience:**

Yes

**Audience Explanation:**

federate learning of llm are in interest of tmlr reader

**Broader Impact Concerns:**

No concerns. The work improves efficiency of federated learning, a privacy-preserving framework, and doesn't introduce new risks.

**Claims And Evidence:**

Yes

**Claims Explanation:**

yes, the claims are supported by data to the best of my knowledge.

**Requested Changes:**

Typo: Lines 13 and 16 of the Reconstruct procedure (algo 1) both update theta_1. Line 16 should update theta_2, and line 13 should use z1 not z.

All current tasks are classification or binary QA. A generative finetuning task (e.g., instruction following or summarization) would broaden the method's demonstrated applicability, especially since the paper frames itself around LLM finetuning.

The paper fixes f2 as only the last linear layer in all experiments with no justification beyond it being "smaller." Varying the split point (e.g., f2 = last 1, 2, 4 transformer layers) and reporting accuracy and computation tradeoffs is essential, as this is arguably the most important design decision in the method. Without this, it is unclear whether the fixed choice is optimal or whether the method is robust to this hyperparameter.

---

> ### Author Response · Authors · 2026-03-23
>
> We thank the reviewer for the review and constructive feedback.
>
> ### Typo
> - Thank you for highlighting the typo; we have adjusted it accordingly.
>
> ### Analyzing Varying split points
> - We analyze the effect of the split point under a fixed perturbation budget on both RoBERTa and OPT models, where earlier splits (e.g., at the 17th, 21st, and 23rd blocks out of the 26 blocks) involve a larger portion of the network in the perturbation process. Each attention(middle) block (24 blocks in total) in these models typically consists of a self-attention module followed by one or two MLP layers in addition to the first and last blocks. The results show that splitting at earlier points leads to higher overall computation, as a larger part of the model must be re-evaluated for each perturbation. From an accuracy perspective, all configurations perform similarly, with only a slight degradation observed for the 20th-block split on MultiRC. These results can be found in Appendix C.
>
>
> ### Broader task
> - To broaden the demonstrated applicability of our method, we have extended our evaluation to include the SQuAD[R1] benchmark. In terms of results, FedSPZO achieves competitive performance compared to FedAvg with LoRA, with a 4.5 percentage point decrease in f1-score, while maintaining similar trends in communication and computation efficiency. Furthermore, compared to zeroth-order baselines such as FedZO and DecomFL, these methods require approximately $1.6\times$ and $2.8\times$ more GFLOPs, respectively, than FedSPZO (which also shows slight f1-score gains). We included these results in the revised version.
>
>
> [R1] Pranav Rajpurkar, Jian Zhang, Konstantin Lopyrev, and Percy Liang. Squad: 100,000+ questions for machine comprehension of text. In Proceedings of the 2016 conference on empirical methods in natural language processing, pp. 2383–2392, 2016

---

### Review · Reviewer_H1Bj · 2026-03-11

**Summary Of Contributions:**

The application of fine-tuning LLMs with federated learning is important. The authors provide a more efficient zeroth-order solution that improves accuracy and convergence compared with existing zeroth-order methods for fine-tuning.

### Strengths
1. The proposed method improves accuracy, convergence, and efficiency, supported by both empirical results and theoretical analysis.
2. The evaluation demonstrates improvements over existing zeroth-order methods.

### Weaknesses
1. Since the authors compare with the baseline FedAvg (LoRA), they may also want to demonstrate that the proposed solution works with LoRA as well.
2. It is somewhat unfair to compare only with SGD as the baseline, since zeroth-order methods are mainly based on gradient estimation. For FedAvg (LoRA), users can also employ other optimizers such as Adam, AdamW, or clipping techniques to significantly improve convergence.
3. The authors should also consider quantization. To reduce memory usage, quantization is commonly introduced during training. The authors may need to demonstrate that the proposed zeroth-order method can work well with quantized models.

### Minor Comments
1. The two algorithms are somewhat difficult to read, especially due to the large number of notations and terminologies. The authors may consider organizing them more clearly. For example, the right side of Figure 1 does not provide much useful information, and the authors may consider improving Figure 1.
2. The authors may also consider evaluating the method on larger models.

**Audience:**

Yes

**Audience Explanation:**

Large language model training, federated learning, and zeroth-order fine-tuning are cutting edge topics and of certain importance to resolve.

**Broader Impact Concerns:**

No ethical implications need to evaluate.

**Claims And Evidence:**

No

**Claims Explanation:**

I think most of the claims are supported by evidence, but there are still some claims that are not completely supported, as listed in the Weakness section.

I also have several questions that the authors may want to discuss or clarify.

1. I am wondering whether the proposed method can be applied to LoRA fine-tuning and other LoRA-based extensions. In particular, the method requires blocks for perturbation. For LoRA matrices, would such perturbations still be appropriate given the different magnitudes typically observed in LoRA parameters?

2. How does the choice of the split position $l$ affect the performance?

3. It would also be beneficial to compare with other optimizers such as AdamW. One drawback of AdamW is its higher memory consumption due to storing additional optimizer states. Since the proposed method aims to reduce memory usage, demonstrating comparable performance to FedAvg (LoRA) with AdamW would make the method particularly compelling.

**Requested Changes:**

1. Experiments to show the method works with LoRA as well
2. Experiments to show the method can beat or comparable to FedAvg (AdamW [Client optimizer and server optimizer])
3. Experiments to show the method can beat or work with training-time quantization algorithms.
4. Authors may check with minor issues.

---

> ### Author Response · Authors · 2026-03-23
>
> We thank the reviewer for the review and constructive feedback.
>
> ### Further comparison with FedAVG(LoRA) with AdamW
> We have included further comparison with FedAVG(LoRA) using AdamW in Appendix A. FedAVG(LoRA) with AdamW performs better compared to FedAVG(LoRA) with SGD at the cost of memory, which can be acceptable when using LoRA but not when fine-tuning the full model parameters.  We discuss this in the appendix and include a comparison between FedAVG(LoRA) using AdamW and FedSPZO. In summary, the same observations of FedSPZO compared to first-order with LoRA methods still hold.
>
> ### LoRA and quantization techniques with FedSPZO
> FedSPZO can work while using LoRA. Furthermore, FedSPZO can also work on top of QLoRA [R1] (quantization with LoRA) for further memory savings. In Appendix B, we show the results on two models and discuss further practical implications. To summarize, FedSPZO is orthogonal and can work with these techniques.
>
> [R1] Tim Dettmers, Artidoro Pagnoni, Ari Holtzman, and Luke Zettlemoyer. Qlora: Efficient finetuning of quantized llms. Advances in neural information processing systems, 36:10088–10115, 2023.
>
> ### Select split point
> In our method, placing the split earlier in the network increases computational overhead, as a larger portion of the model must be re-evaluated for each perturbation, without yielding noticeable gains in convergence speed. This behavior is demonstrated in Appendix C for the evaluated model architectures. From an accuracy perspective, model performance remains largely consistent across different split choices, indicating that the method is not highly sensitive to this parameter.
>
> ### We further improve the algorithm presentation for easier readability (including simplifying and reorganizing the seed-related lines in Algorithm 2) and added more descriptive annotations.

---

> > ### Comment · Reviewer_H1Bj · 2026-04-03
> >
> > Thank the authors for their reply. I apologize for the delay in my response.
> >
> > I have a further question regarding the choice of the split point. The experiments in Appendix C are very insightful, but I am wondering whether there is a general rule for selecting the split point. For example, should one use a fixed layer number or a fixed percentile of layers?
> >
> > In Appendix C, layers 17 and 21 appear to be good choices for OPT-1.3B. If I instead use a different model, such as LLaMA-8B or Qwen-32B, should I use the same layer indices or the same percentile? How should the split point be determined in general? Would it be necessary to repeat the experiments in Appendix C for each new model?

---

> ### Author Response · Authors · 2026-04-04
>
> We thank the reviewer for the response. We would like to clarify that our results in Appendix C show that later split points (i.e., closer to the end of the network) consistently achieve the best trade-off, while earlier splits (e.g., layers 17 or 21 in RoBERTa-large(355M) or OPT-1.3B) incur higher computational cost without providing accuracy benefits. The general and practical rule is to place the split point at the last untied linear layer or the last attention block, which are robust and reliable choices.
>
> Importantly, this guideline does not depend on fixed layer indices (e.g., selecting specific layers such as 17 or 21), but rather on the relative position in the network, namely choosing layers close to the end, where they represent a smaller fraction of the total computation under our split formulation. As a result, for the other models, we recommend selecting the split at the last untied linear layer or  transformer block. Consequently, there is no need to conduct the analysis in Appendix C for every model; choosing a split point in the final layers provides a reliable and efficient default in practice.
>
> We are happy to provide further clarification if needed.

---

### Review · Reviewer_1pFV · 2026-03-12

**Summary Of Contributions:**

This paper proposes FedSPZO, a zeroth-order federated learning method that splits the model into two blocks, aiming to balance between computation, memory and communication efficiency. While the idea of leveraging model structure for more efficient ZO gradient estimation is interesting, the paper suffers from fundamental issues in practice and algorithmic presentation.

**Audience:**

Yes

**Audience Explanation:**

The paper would interest members of the TMLR audience working on federated learning and LLM fine-tuning.

**Claims And Evidence:**

Yes

**Claims Explanation:**

The core claims are supported by experimental results. The experimental results provided in the paper are good, but I have additional concerns regarding the practical use cases of the proposed algorithm. The algorithm description can be improved.

**Requested Changes:**

* I think the proposed method has significant limitations, as it is hard to generalize to generation tasks. Currently, all experiments evaluate only classification tasks. One major concern is that for generative models, the last linear layer is the language modeling head with vocabulary projection. This makes the overhead of the proposed method no longer negligible, and the entire efficiency benefit collapses. Thus, performance on complex tasks like instruction following, dialogue, and reasoning remains a fundamental limitation.

* The authors mentioned that the proposed FedSPZO achieves a better trade-off between memory, communication, and computation overhead. I think this is not clear, as the "better trade-off" is vague. In my view, loading a LLaMA-3.2-3B alone requires ~6GB of memory, and the memory gap between FedSPZO and LoRA-based fine-tuning is only ~3GB; the difference in communication is not that significant as well. However, FedSPZO requires 17×higher total computation than FedAvg(LoRA) and sometimes suffers from performance drops.

* The algorithm description is not very well organized and is quite hard to follow. For example, $sl_2$ is passed as a parameter to Forward, but it is not an output, nor is the output clearly related to $sl_2$. However, in Line 15, $sl_2$ is further used. It is unclear from the pseudocode whether $sl_2$ serves as an input, an output, or both.

* The paper has issues with citation formatting throughout. The authors frequently misuse \citet and \citep, placing citations (including author names and years) directly between a concept and its predicate.

* Some questions related to the experimental settings:
  * Why were FedAvg and FedAvg(LoRA) evaluated using vanilla SGD without optimizer states?
  * What is the reason behind using different batch sizes when evaluating different methods?

---

> ### Author Response · Authors · 2026-03-23
>
> We thank the reviewer for the review and constructive feedback.
>
> ### One major concern is that for generative models, the last linear layer is the language modeling head with vocabulary projection. This makes the overhead of the proposed method no longer negligible, and the entire efficiency benefit collapses
>
> We thank the reviewer for raising this point. In all our experiments, we adopt task-aligned prompt-based fine-tuning [R1], where downstream tasks are reformulated to match the pretraining objective. Under this setting, the original language modeling head with the vocabulary projection is retained and used for token prediction without introducing any additional task-specific output layer. The model is therefore fine-tuned using the same architecture as in pretraining, including the existing LM head. Therefore, the large vocabulary projection layer is part of the standard model architecture used during fine-tuning, and all reported results in the paper are obtained under this setting. Consequently, the efficiency improvements of FedSPZO are evaluated with the same generative model architecture, including the same language modeling head. In addition, we included further evaluation on the generative task of SQuAD[R2], where the gains still persist.
>
> We also note that the use of task-aligned fine-tuning was mentioned in the abstract, the related work section (with citation), and the experimental setup (section 4.1). To avoid any potential ambiguity, we have added the following clarification in the experimental setup:
>
> To make this clearer, we have added the following clarification to the manuscript in the experimental setup section:
> We evaluate FedSPZO alongside other FL approaches with prompt-based **(task-aligned)** tuning **(Gao et al.,
> 2021), where downstream tasks are reformulated to match the pretraining objective without introducing a
> task-specific output layer while retaining the pretrained language modeling head.**
>
> [R1] Tianyu Gao, Adam Fisch, and Danqi Chen. Making pre-trained language models better few-shot learners. In Chengqing Zong, Fei Xia, Wenjie Li, and Roberto Navigli (eds.), Proceedings of the 59th Annual Meeting of the Association for Computational Linguistics and the 11th International Joint Conference on Natural Language Processing (Volume 1: Long Papers). Association for Computational Linguistics, August 2021.
>
> [R2] Pranav Rajpurkar, Jian Zhang, Konstantin Lopyrev, and Percy Liang. Squad: 100,000+ questions for machine comprehension of text. In Proceedings of the 2016 conference on empirical methods in natural language processing, pp. 2383–2392, 2016

---

> > ### Author Response · Authors · 2026-03-23
> >
> > ### The authors mentioned that the proposed FedSPZO achieves a better trade-off between memory, communication, and computation overhead. I think this is not clear, as the "better trade-off" is vague. In my view, loading a LLaMA-3.2-3B alone requires ~6GB of memory, and the memory gap between FedSPZO and LoRA-based fine-tuning is only ~3GB
> >
> > For the phrase “better trade-off”, our intention is to highlight that FedSPZO balances memory, communication, and computation differently compared to existing approaches. In particular, FedSPZO significantly reduces memory consumption compared to first-order fine-tuning methods, including LoRA-based approaches. While the reviewer notes that the difference can be around 3–4 GB for first-order LoRA, this gap is practically important in resource-constrained edge environments. For instance, loading a model such as OPT or LLaMA already requires roughly 6 GB of memory. On devices commonly used in federated learning scenarios—such as high-end smartphones (e.g., iPhone) or embedded platforms like the NVIDIA Jetson Orin Nano, which typically provide 8 GB of RAM, this additional 3-4 GB determines whether training is feasible. In contrast, our approach enables training within these memory limits. Moreover, this memory advantage becomes more pronounced as the input context length increases since first-order methods must store larger activations for backpropagation.
> >
> > Besides, our approach substantially reduces communication overhead, typically by three orders of magnitude compared to parameter-efficient federated learning methods. Note that both computation and communication overhead can increase convergence time, although computation typically plays a more important role in convergence, as discussed in the paper. Communication, however, has an additional practical implication: it is performed over resources leased from a third party (e.g., a wireless operator). Therefore, optimizing communication can be even more important when this leased resource is costly, as is often the case for wireless links. Our solution can therefore serve as an alternative to LoRA when communication is the bottleneck or the more costly resource. As a summary, our method provides a practical trade-off: it substantially reduces memory usage (often the hard constraint) and significantly lowers communication overhead while accepting a higher computational cost compared to first-order methods, which is an improvement relative to existing zeroth-order approaches.
> >
> > To make that part clearer, we have revised the sentence in the manuscript (highlighted in blue) ".., which reduces the computational cost compared to state-of-the-art federated zeroth-order approaches in finetuning language models. At the same time, it retains the key advantages of zeroth-order optimization, including improved memory efficiency and reduced communication overhead compared to federated first-order methods."
> >
> >
> > ### Organizing algorithm presentation
> >
> > - We understand the confusion that might occur from this use of the seeds list $sl_2$. Therefore, we adjust this use of $sl_2$, where the seeds per step are sampled and appended to $sl_2$ (in Line 9) and used later in the two forward calls (where the seed lists are just passed as input). We also added more annotations to improve the presentation.
> >
> > ### citation formatting
> > - We have revised the citation formatting in the text.
> >
> > ### Why no optimizer states
> >
> > We did not include full-parameter FedAvg with adaptive  optimizers since they introduce substantial memory overhead. In particular, Adam/AdamW maintains first- and second-moment estimates for each parameter, resulting in an optimizer state of approximately $2\times$ the trainable parameter size on top of vanilla first-order SGD. We followed the same optimizer with LoRA; however, we also added a further comparison using AdamW in Appendix A.
> >
> > ### Why different batch sizes.
> > In general, the intention of this comparison was to illustrate that the larger memory footprint of first-order methods compared to FedSPZO cannot simply be mitigated by reducing the batch size or applying simple implementation tricks. Therefore, the specific batch size shown in the figure is not a critical aspect of the comparison, but rather serves to highlight the relative memory efficiency of FedSPZO.

---

> > > ### Comment · Reviewer_1pFV · 2026-04-06
> > >
> > > Thank the authors for the rebuttal, and apologize for the delay. I believe the authors have addressed all my concerns.

---

### Decision · Action_Editor_HtnC · 2026-05-29

**Recommendation:** Accept as is

**Audience:**

Yes

**Audience Explanation:**

Federated learning and the efficient fine-tuning of Large Language Models (LLMs) on edge devices are highly active and critical areas of research. The TMLR audience will find the proposed zeroth-order optimization approach highly relevant and practically useful. All reviewers universally agreed on the strong audience interest for this work.

**Claims And Evidence:**

Yes

**Claims Explanation:**

The core claims regarding the efficiency of the proposed FedSPZO method are well-supported by both empirical results and the authors' detailed rebuttal. Initially, reviewers raised valid questions regarding the method's applicability to generative tasks, compatibility with standard optimizers (like AdamW) and quantization techniques (QLoRA), and the justification for the fixed split point. During the rebuttal, the authors give more details to address these concerns.